# Mechanical Behavior Analysis of Neural Electrode Arrays Implantation in Brain Tissue

**DOI:** 10.3390/mi16091010

**Published:** 2025-08-31

**Authors:** Xinyue Tan, Bei Tong, Kunyang Zhang, Changmao Ni, Dengfei Yang, Zhaolong Gao, Yuzhao Huang, Na Yao, Li Huang

**Affiliations:** 1Wuhan Neuracom Technology Development Co., Ltd., Wuhan 430023, China; tanxinyue@neuracom.com.cn (X.T.); tongbei@neuracom.com.cn (B.T.); nichangmao@neuracom.com.cn (C.N.);; 2Mechanical Science and Engineering, Huazhong University of Science and Technology, Wuhan 430074, China

**Keywords:** BCI, neural electrode arrays, mechanical behavior, finite element, insertion force

## Abstract

Understanding the mechanical behavior of implanted neural electrode arrays is crucial for BCI development, which is the foundation for ensuring surgical safety, implantation precision, and evaluating electrode efficacy and long-term stability. Therefore, a reliable FE models are effective in reducing animal experiments and are essential for a deeper understanding of the mechanics of the implantation process. This study established a novel finite element model to simulate neural electrode implantation into brain tissue, specifically characterizing the nonlinear mechanical responses of brain tissue. Synchronized electrode implantation experiments were conducted using ex vivo porcine brain tissue. The results demonstrate that the model accurately reproduces the dynamics of the electrode implantation process. Quantitative analysis reveals that the implantation force exhibits a positive correlation with insertion depth, the average implantation force per electrode within a multi-electrode array decreases with increasing electrode number, and elevation in electrode size, shank spacing, and insertion speed each contribute to a systematic increase in insertion force. This study provides a reliable simulation tool and in-depth mechanistic analysis for predicting the implantation forces of high-density neural electrode arrays and offer theoretical guidance for optimizing BCI implantation device design.

## 1. Introduction

Brain-computer interfaces (BCIs) represent a transformative class of bioelectronic systems designed to interface directly with neural circuitry for restoring sensorimotor functions or augmenting central nervous system capabilities [1,2]. Central to BCI implementation, neural electrode technology enables two primary functionalities: high-fidelity electrophysiological recording from targeted brain regions to investigate neuronal population dynamics, and precision electrical stimulation for neuro-modulation in neurological disorder management [3]. Contemporary commercial neural electrodes predominantly exist in two architectural paradigms: Utah-style arrays and Michigan-type probes [4,5]. The Utah electrode configuration, originally developed at the University of Utah, employs a planar microneedle array with discrete recording sites localized at individual probe tips. While this design facilitates stable cortical surface interfacing, its spatial sampling resolution is fundamentally constrained by channel density limitations, rendering it particularly suitable for superficial-layer neural interfacing. In contrast, the Michigan electrode architecture, fabricated via micro-electro-mechanical systems (MEMS)-based microfabrication techniques, incorporates multiple recording sites along each penetrating shank. This configuration enables high-density neural sampling with enhanced spatial resolution while maintaining minimal tissue displacement, attributed to its subcellular-scale structural dimensions. Given these comparative advantages in spatial resolution and tissue compatibility, the Michigan electrode architecture was selected as the computational model for this investigation of neural interface implantation mechanics.

Neural electrode implantation presents a critical biomechanical challenge requiring a precise balance between structural integrity and tissue preservation. Successful long-term signal acquisition necessitates a comprehensive evaluation of both mechanical reliability and chronic biocompatibility. From tissue mechanics perspectives, ideal implantation must satisfy three criteria: (1) maintenance of electrode structural stability without buckling/fracture, (2) minimization of induced stress-strain fields in parenchyma, and (3) preservation of neurovascular architecture for sustained electrophysiological functionality. Prior experimental studies have systematically characterized implantation mechanics across various model systems. C. Bergaud et al. [6] demonstrated biodegradable substrate electrodes optimized through murine implantation studies, achieving improved stress distribution profiles. Bjornsson et al. [7] established correlations between electrode tip geometry/insertion velocity and tissue deformation metrics in ex vivo rat brains, with particular emphasis on vascular damage mitigation. Casanova et al. [8] quantified frictional forces during rodent brain insertions as a key biomarker for tissue-device interaction. Ulbert’s pioneering work [9] on silicon electrode penetration through rat dura mater introduced sharp-tip modification techniques to reduce tissue resistance. Geramifard et al. [10] further developed amorphous silicon carbide probes through cortical insertion experiments, identifying that reduced insertion velocities decrease peak insertion forces in single-shank configurations. They conducted experiments on the implantation into the rat cortex, concluding that slower implantation speeds can reduce the maximum insertion force of a single-shank needle without focusing on the deformation of the rat cortex during implantation.

However, existing studies face significant challenges in accurately measuring brain tissue deformation and stress distribution. Investigating both implantation forces and nonlinear tissue deformation remains crucial for advancing neural interface technology. Computational visualization of the electrode implantation process enables simulation of strain field development in brain tissue, which can optimize electrode dimensions and structural configurations to facilitate reliable implantation while minimizing structural failure and tissue damage. Current simulation-based research on neural electrode-brain interactions predominantly focuses on post-implantation cortical interfaces, particularly addressing chronic tissue/electrode damage caused by micromotion-induced electrode shearing [11,12,13,14,15,16,17]. Few finite element studies have systematically analyzed the actual cortical implantation process, and existing attempts exhibit critical limitations. This challenge arises from the brain tissue’s propensity for extreme mechanical deformations during insertion, which complicates finite element modeling. In one approach, Singh et al. [18] implemented a deformable Lagrangian algorithm to simulate microneedle penetration into brain-mimetic gels and rat brain tissue, utilizing element deletion to achieve substrate penetration. Through iterative adjustment of element deletion criteria, their model achieved temporal force profile alignment with experimental measurements. However, this methodology artificially removes elements exceeding strain thresholds rather than simulating physiological tissue displacement patterns, resulting in non-physiological failure mechanisms. Furthermore, the empirical calibration of deletion thresholds introduces methodological inconsistencies. Another study by Li et al. [19] combined experimental and computational techniques to analyze crack propagation patterns in gels during electrode insertion with varying tip geometries. While successfully characterizing initial crack formation mechanisms, their model failed to simulate complete insertion trajectories beyond superficial depths. O’Sullivan et al. [20] employed an Eulerian algorithm to model four-shank electrode array implantation, effectively capturing large tissue deformations for damage assessment. Although demonstrating velocity-dependent implantation forces and structural influences on tissue damage, the absence of experimental validation undermines model reliability, particularly regarding vascular rupture predictions.

This study establishes an advanced finite element (FE) modeling framework to simulate the mechanical complexity of neural electrode implantation in cerebral tissue, systematically investigating both single-shank electrodes and multi-shank arrays (8- and 32-shank configurations). The developed model achieves dual fidelity by precisely replicating brain tissue biomechanical properties while characterizing dynamic tissue-electrode interactions, specifically addressing nonlinear deformation mechanisms during insertion. To ensure model validity, we implemented an integrated computational-experimental validation protocol through parallel simulations and physical implantation trials using porcine brain specimens. Key findings demonstrate a positive correlation between implantation force magnitude and insertion depth, and an inverse relationship between per-shank implantation force and array density in multi-electrode configurations. The strong agreement between experimental measurements and computational predictions validates the model’s predictive accuracy for both transient insertion dynamics and chronic interface mechanics.

## 2. Materials and Methods

### 2.1. Fabrication of Neural Electrode

In this study, in reference to other bioelectrode manufacturing processes [21,22], the neural electrodes were fabricated by the MEMS process. Silicon (Si) was selected as the substrate material for the neural electrodes and aluminum (Al) for the metal leads, with gold (Au) and titanium nitride (TiN) serving as the materials for the electrode contact points. Initially, a 0.4 micrometer (μm) thick layer of aluminum metal was deposited on an 8-inch silicon wafer substrate using Physical Vapor Deposition (PVD) technology. Subsequently, photolithography and dry etching were employed to pattern the metal leads. Following this, a 0.3 μm thick layer of silicon nitride (SiNx) insulation was deposited using Plasma-Enhanced Chemical Vapor Deposition (PECVD). Subsequently, plasma etching was used to remove the silicon nitride layer from the pad area, and a conductive seed layer consisting of 30 nm of titanium (Ti) and 100 nm of copper (Cu) was deposited for the electroplating process. And then the process of photoresist coating, exposure, and development was repeated to expose the contact areas of the electrodes. Following this, a 50 nm titanium nitride (TiN) functional modification layer was deposited via physical vapor deposition (PVD).

As shown in Figure 1, the schematic depicts single-shank neural electrodes and multi-shank arrays (8- and 32-shank configurations). The single-shank electrode measures 5.5 mm (length) × 100 μm (width) × 70 μm (thickness), with a 30° beveled tip. For multi-shank arrays, the center-to-center inter-shank spacing is 230 μm. These electrodes served as experimental samples for implantation trials in this study.

### 2.2. Experimental Equipment and Method

Due to the biomechanically comparable properties between porcine and human brain tissues [23,24], fresh porcine brains were selected as implantation substrates. Brains were procured from a certified slaughterhouse on the day of experimental use and maintained in phosphate-buffered saline (PBS)-filled Petri dishes at 37 °C. The meninges were carefully removed prior to the experiment. Prior to each trial, tissues were rehydrated with PBS to preserve native mechanical characteristics.

Figure 2 illustrates the custom-built three-axis microforce measurement system for quantifying implantation dynamics. The apparatus integrates: (1) a piezoresistive force sensor (Nators, Suzhou, China; 0.5 μN resolution, 0–200 mN range), (2) a linear displacement stage (10 nm positioning accuracy, 8 mm travel range), and (3) data acquisition modules (50 Hz sampling rate). Force-displacement synchronization was achieved through PC-controlled instrumentation.

Neural electrodes (single-shank and 8/32-shank arrays) were fixed to the force sensor probe via biocompatible adhesive. Controlled insertions were performed at 1 mm/s velocity to 1 mm depth in cortical regions. Real-time force-displacement profiles were recorded throughout implantation cycles.

This methodology enables precise characterization of electrode-tissue interaction mechanics, capturing critical implantation parameters under physiologically relevant conditions. The combination of fresh tissue preparation and anatomically preserved implantation sites ensures clinical fidelity, while standardized protocols enhance experimental reproducibility.

### 2.3. Simulation

Finite element modeling was performed using ABAQUS/Explicit 2019 (Dassault Systemes, Vélizy-Villacoublay, France) with the coupled Eulerian-Lagrangian (CEL) method to simulate neural electrode implantation. Three distinct models were developed: single-shank electrodes and multi-shank arrays (8-shank and 32-shank configurations), as illustrated in Figure 3. Multi-shank arrays employed half-symmetry modeling to reduce computational costs. Electrodes were modeled as Lagrangian solids with dimensions matching physical prototypes, while brain tissue occupied the Eulerian domain. A uniform mesh size of 0.02 mm (1/5 electrode width) was applied to both components after mesh convergence verification. The Eulerian domain extended 10× the electrode width horizontally to minimize boundary effects. Model parameters are summarized in Table 1, with mesh statistics detailed in Table 2.

Electrodes received the prescribed *Z*-axis displacement (1 mm/s) with other degrees of freedom constrained. Brain tissue boundaries were fixed at the base with restricted vertical motion. The Eulerian formulation naturally captured surface deformation by incorporating both brain tissue and air layers during insertion simulations.

Brain tissue is extremely soft and undergoes large deformations during electrode insertion. In line with studies on its material properties [23,24,25,26,27,28,29,30], we employed a hyperelastic constitutive model to accurately represent this behavior. Material parameters were calibrated through published experimental data [24], specifically adopting the Ogden hyperelastic framework. Following literature-derived parameters, the brain tissue’s mechanical response was modeled using a first-order Ogden strain energy density function:(1)$$W=\frac{2\mu }{\alpha^{2}}(\lambda_{1}^{\alpha }+\lambda_{2}^{\alpha }+\lambda_{3}^{\alpha }-3)$$
where the *λ_i_* represents principal stretch ratios (defined as the square roots of eigenvalues (C), with C being the right Cauchy-Green strain tensor). Material-specific coefficients *μ* and *α* govern hyperelastic stiffening behavior. The nominal stress-strain relationship is derived through partial differentiation of the strain energy function:(2)$$S_{11}=\frac{2\mu }{\alpha }[\lambda^{\alpha -1}-\lambda^{-(\frac{\alpha }{2}+1)}]$$

For simulation analyses, porcine brain tissue material properties were defined per Table 3. Given that the silicon-based electrode’s elastic modulus (~160 GPa) exceeds brain tissue stiffness by six orders of magnitude, electrodes were modeled as rigid bodies consistent with prior methodologies.

Material parameter implementation in finite element frameworks requires rigorous verification of constitutive model integration accuracy. The adopted hyperelastic formulation enables precise modeling of the brain tissue’s nonlinear mechanical response during electrode insertion.

To gain a comprehensive understanding of the factors influencing microelectrode insertion forces, we conducted a systematic investigation into the effects of shank thickness, shank spacing in multi-shank arrays, and insertion speed. While limitations in electrode fabrication cost and the measurement range of experimental equipment prevented exhaustive experimental study, the FE model proposed in this study reliably complemented the experimental work. Building upon the baseline simulation model—which has been validated against experimental data—additional FEA cases were established as summarized in Table 4. All settings not varied in these cases remained consistent with the baseline model.

## 3. Results

### 3.1. Implantation Experiment Results

Figure 4 displays experimental implantation sequences and corresponding force-displacement profiles for three electrode configurations. The horizontal axis denotes insertion depth (0–1 mm) at 1 mm/s velocity, while the vertical axis quantifies required implantation force. All profiles exhibit monotonic force escalation with depth progression. Mean implantation forces (±SD, n = 4 trials) measured at 1 mm depth were Single-shank: 538.6 ± 94.55 μN; 8-shank array: 1879.4 ± 505.97 μN; 32-shank array: 4792.7 ± 1444.86 μN.

Critical analysis reveals an inverse relationship between per-shank implantation force and array density (Table 5), demonstrating force distribution advantages in high-density configurations. These empirical correlations between electrode geometry and mechanical loading provide essential guidelines for surgical safety optimization through force prediction, electrode structural design refinement, and chronic interface stability enhancement in clinical neurotechnology applications.

### 3.2. Simulation Results of Different Shank Numbers

Figure 5 illustrates deformation profiles during 1 mm implantations of single-shank and 8/32-shank neural electrodes, alongside finite element analysis (FEA)-experimental force-displacement correlations. Tissue deformation patterns demonstrate microneedle-induced indentation, aligning with physical insertion dynamics.

FEA predictions were benchmarked against experimental data for each configuration. Simulated force profiles exhibited strong concordance with measured values across all electrode densities, validating model accuracy for both single-shank and multi-shank array simulations. Multi-shank symmetry modeling preserved full-structure mechanical behavior while reducing computational costs.

The stress in brain tissue (Figure 6) and its contact pressure on neural electrode shanks (Figure 7) were investigated when electrodes with different shank numbers were implanted to a depth of 1 mm. Notably, as the number of electrode shanks increases, the contact pressure on a single shank decreases, directly leading to lower resistance per shank in the electrode array. Von Mises stress calculations in brain tissue revealed that for a single-electrode implant, stress values over 3 kPa were generated around the non-inserted part of the brain tissue. In contrast, for 8-shank and 32-shank array-type electrodes, very small stresses (less than 1 kPa) were observed in the rest of the electrodes except the tip, especially between two electrodes. In the depth direction, 32-shank electrodes had a smaller stress-influence range than 8-shank electrodes. Calculations of interfacial contact pressure and internal brain tissue stresses show that densely arranged multi-shank electrodes significantly restrict the local brain tissue, reducing the compression of brain tissue on the electrodes and thus the average implantation force on each electrode shank.

### 3.3. Simulation Results of Different Dimensions

Taking the single-shank electrode as an example, we investigated the insertion forces of electrodes with different dimensions based on simulation results. As shown in Figure 8, the trend of insertion force versus depth is presented for shank thicknesses of 50 μm, 70 μm, and 90 μm. The results indicate that while the general trend of the insertion force curves remains consistent across different thicknesses, the magnitude of the force increases with greater shank thickness.

Using an 8-shank array as an example, we further investigated the insertion forces of arrays with different center-to-center inter-shank spacing (0.23 mm, 0.46 mm, and 0.69 mm). The results shown in Figure 9 indicate that the insertion force increases with larger inter-shank spacing, which can be attributed to reduced mechanical constraint between shanks and greater localized tissue deformation.

### 3.4. Simulation Results of Different Insertion Speeds

To assess the impact of insertion speed on the implantation force, the relationship between force and depth was further examined during insertions to a depth of 1 mm at speeds of 0.1 mm/s and 10 mm/s. As shown in Figure 10, under the 0.1 mm/s condition, experimental results for single-shank, 8-shank, and 32-shank array insertions are presented alongside a comparison with simulation results derived from the model introduced in Section 3.2. The computational predictions continue to demonstrate strong agreement with experimental measurements at this speed, confirming that the proposed model retains its reliability when applied to other insertion velocities.

Figure 11 presents the simulated insertion force versus depth at an insertion speed of 10 mm/s. Experimental validation at this speed was not feasible due to the maximum speed limitation of the force-displacement sensor. Despite this, the reliability of the simulation model was confirmed under both 0.1 mm/s and 1 mm/s conditions.

Figure 12 compares the insertion forces at 1 mm depth for different electrode configurations and insertion speeds. As the insertion speed increases, the required insertion force rises for all electrode types, which can be attributed to the strain-rate-dependent stiffening behavior of brain tissue. Furthermore, the previously observed trend—where the average insertion force per shank decreases as the number of shanks increases—remains at the 1 mm/s insertion speed.

## 4. Discussion

Existing experimental studies [7,9,10] focusing on neural electrode implantation in rodent models predominantly employ force-displacement measurements and post-operative histological analysis. While these methods provide foundational biomechanical data, they face dual constraints of high experimental costs and limited capacity for three-dimensional damage visualization. Although prior finite element investigations have simulated probe penetration through epidermal tissue, fundamental differences in mechanical properties between dermal and cerebral tissues preclude direct translation of such models. This work systematically addresses these limitations through an integrated experimental-computational framework specifically validated for cerebral electrode implantation mechanics.

To accurately characterize the severe deformation behavior of the cerebral cortex, we used the CEL method to model the electrode implantation process. Force-deformation patterns and established bio-mechanical evidence confirm the hyperelastic model as optimal for simulating porcine brain tissue dynamics. Material parameters were calibrated using published experimental datasets [23,24,25,26,27,28,29]. Given the impracticality of direct strain-rate measurement during array implantation, experimental-simulated force curve comparison provides effective parameter optimization and model validation.

The insertion resistance of neural electrodes in brain tissue primarily comprises two components: the compressive force exerted by the electrode tip on the tissue, and the frictional resistance generated by tissue encapsulation around the electrode shaft [30]. The latter exhibits a direct positive correlation with the degree of tissue encapsulation. During single-electrode implantation, significant cortical surface collapse occurs, causing deformed tissue to encapsulate the electrode and generate insertion resistance. In multi-shank electrode implantation, although cortical collapse persists, densely distributed electrodes collectively distribute this encapsulation force. As shown in the brain tissue stress contour plot (Figure 7), stress values in interstitial regions between multi-shank electrodes are lower than those surrounding single electrodes, indicating reduced tissue encapsulation. Furthermore, centrally located electrodes experience diminished compressive forces and correspondingly reduced contact friction, ultimately resulting in lower average insertion force per electrode.

The increase in insertion force resulting from greater electrode thickness can be attributed to two main factors: the elevated pushing force due to the increased cross-sectional area, and the heightened friction force caused by the larger lateral surface area. This mechanism has been corroborated by existing experimental studies on insertion forces [15,17].

The total insertion force of 8-shank electrode arrays rises with larger inter-shank spacing, consistent with the principle that single-shank insertion force exceeds the average per-shank force of 8-shank. As the shank spacing increases, the mechanical conditions for each electrode increasingly resemble those of a single electrode. The brain tissue between shanks exerts greater resistance on each electrode, leading to an overall increase in insertion force. Furthermore, as illustrated by the cortical surface deformation patterns in Figure 9A–C, electrodes with wider spacing are required to push a larger area of tissue, which also contributes significantly to the increase in insertion force.

Brain tissue, characterized by its Ogden hyperelastic material properties, exhibits a distinct strain-rate-dependent stiffening response—a behavior well-documented in the literature on cerebral biomechanics [25,26,27,29]. As the insertion speed increases, the strain rate in the tissue rises correspondingly, resulting in increased stiffness. This enhanced stiffness imposes greater resistance on the electrode, thereby leading to a higher insertion force. Observations of cortical surface displacement during insertions to a depth of 1 mm further reveal that at the low speed of 0.1 mm/s, no significant surface push-in occurs. This can be attributed to the markedly reduced stiffness of brain tissue under low strain-rate conditions, coupled with the extended time available for tissue redistribution around the shank. Together, these factors contribute to the diminished surface deformation observed during slow insertions.

Notably, although the average insertion force per shank decreases with increasing number of electrodes in multi-shank arrays, this does not imply lower implantation failure risk compared to single-shank electrodes [31,32,33,34,35]. Once the electrode tip penetrates brain tissue, its degrees of freedom perpendicular to the electrode axis become constrained. During multi-shank electrode implantation, edge-positioned electrodes encounter an effective oblique insertion plane formed by cortical surface collapse within their coverage area. The height differential across this collapsed region induces lateral forces directed toward the array center, predisposing these peripheral electrodes to fracture.

Our methodology underscores the critical importance of precise modeling and experimental validation for understanding complex electrode-tissue interactions. Finite element analysis enables in-depth analysis of mechanical behaviors during implantation, which is essential for electrode design optimization and tissue damage minimization. The fidelity between simulation and experimental results reinforces the reliability of the finite element model, establishing a foundation for future neural electrode array research and large-scale neuronal signal acquisition applications.

## 5. Conclusions

This study developed a validated FE model using the CEL method to simulate the complex dynamics of neural electrode implantation into brain tissue, specifically characterizing the bio-mechanical interactions and nonlinear tissue deformation mechanisms.

Experiments involving the implantation of single-shank and multi-shank electrodes into ex vivo porcine brain tissue demonstrated consistency between simulation results and actual measurements, thereby validating the model’s accuracy. The results indicate a positive correlation between neural electrode insertion force and implantation depth. Furthermore, the average insertion force per electrode within a multi-electrode array decreases as the number of electrodes increases, and the insertion force increases with electrode size, shank spacing, and insertion speed.

By providing a predictive and experimentally validated simulation framework, this study could help to reduce the reliance on large-scale animal experiments while maintaining scientific rigor. This study provides a basis for predicting the insertion forces of high-density neural electrode arrays and offers significant guidance for optimizing the structural design parameters of electrodes and the design of BCI implantation devices.

## Figures and Tables

**Figure 1 micromachines-16-01010-f001:**
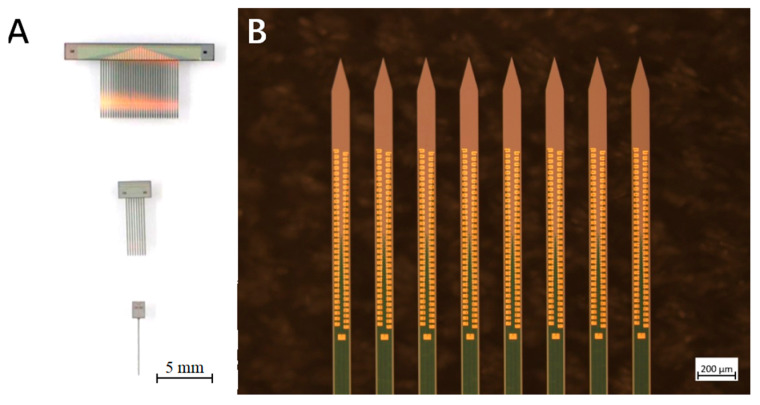
(**A**) From bottom to top: single-shank neural electrode, 8-shank neural electrode, and 32-shank neural electrode; (**B**) 8-shank neural electrode SEM.

**Figure 2 micromachines-16-01010-f002:**
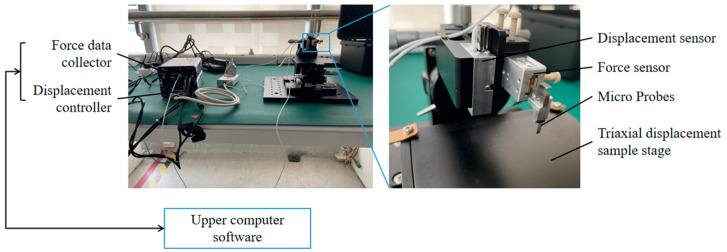
Micro-force Sensor System.

**Figure 3 micromachines-16-01010-f003:**
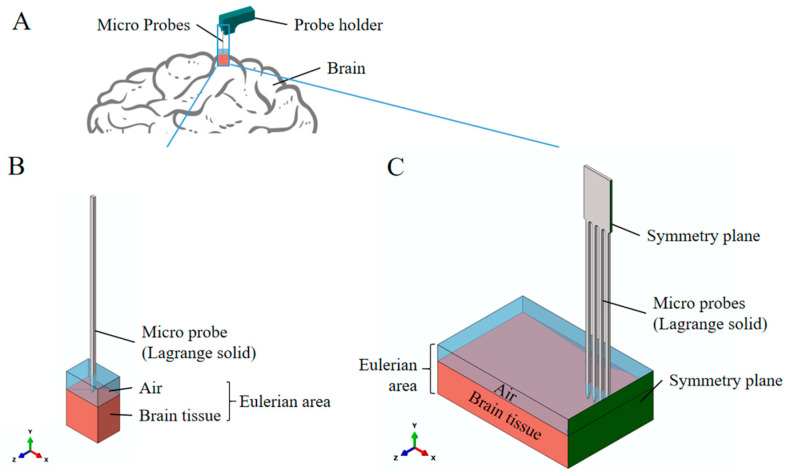
(**A**) Modeling Schematic; (**B**) Single-shank neural electrodes model; (**C**) 8-shank neural electrodes model.

**Figure 4 micromachines-16-01010-f004:**
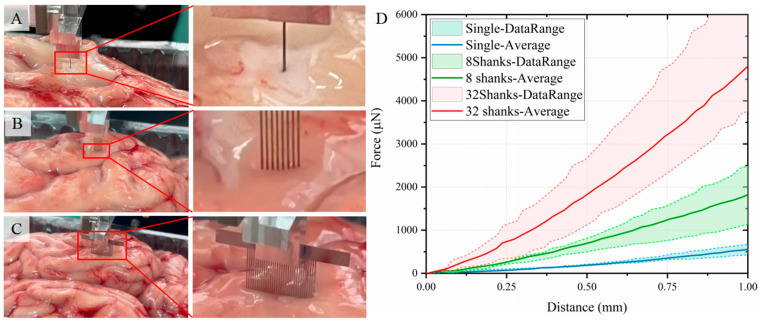
Mechanical characterization of neural electrode implantation in porcine brain tissue at 1 mm/s velocity and 1 mm target depth. (**A**–**C**) Sequential implantation images of single-shank, 8-shank, and 32-shank electrode arrays, showing tissue indentation near electrode interfaces (insets: magnified views). (**D**) Experimental force-displacement profiles (n = 4 trials) and mean curves for each configuration. Dashed lines indicate max/min values, shaded areas show data distribution ranges, and solid lines represent mean values. Results show implantation force increases with depth. Mean forces (±SD) at 1 mm depth: 538.6 ± 94.55 μN (single-shank), 1879.4 ± 505.97 μN (8-shank), 4792.7 ± 1444.86 μN (32-shank).

**Figure 5 micromachines-16-01010-f005:**
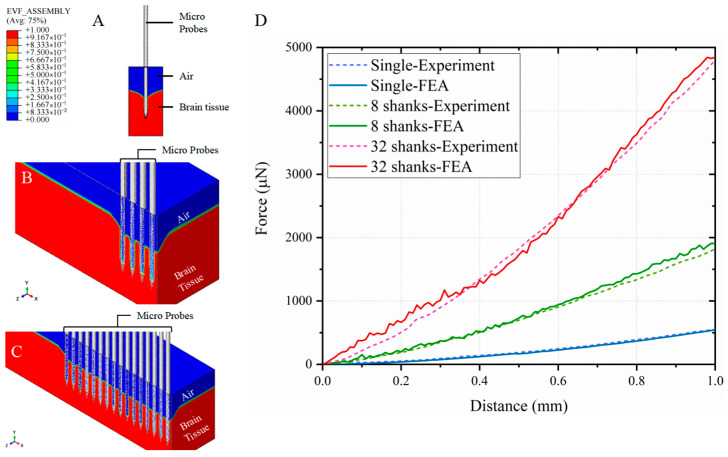
Computational modeling of neural electrode implantation (1 mm/s velocity, 1 mm depth) in porcine brain tissue. Simulated cross-sectional views of (**A**) single-shank, (**B**) 8-shank, and (**C**) 32-shank electrode insertions, demonstrating cortical surface indentation patterns consistent with experimental observations (cf. Figure 4). (**D**) Comparative force-depth profiles between computational predictions (solid lines) and experimental means (dashed lines) configurations show that experimental and simulation results closely align.

**Figure 6 micromachines-16-01010-f006:**
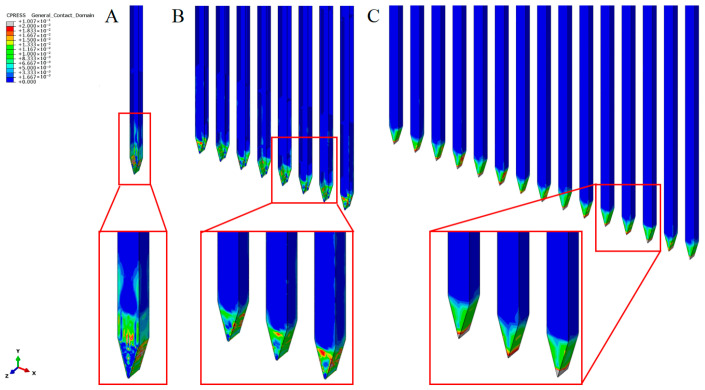
Contact pressure at the electrode-brain tissue surface at 1 mm implantation depth. (**A**) Single shank; (**B**) 8 shanks; (**C**) 32 shanks.

**Figure 7 micromachines-16-01010-f007:**
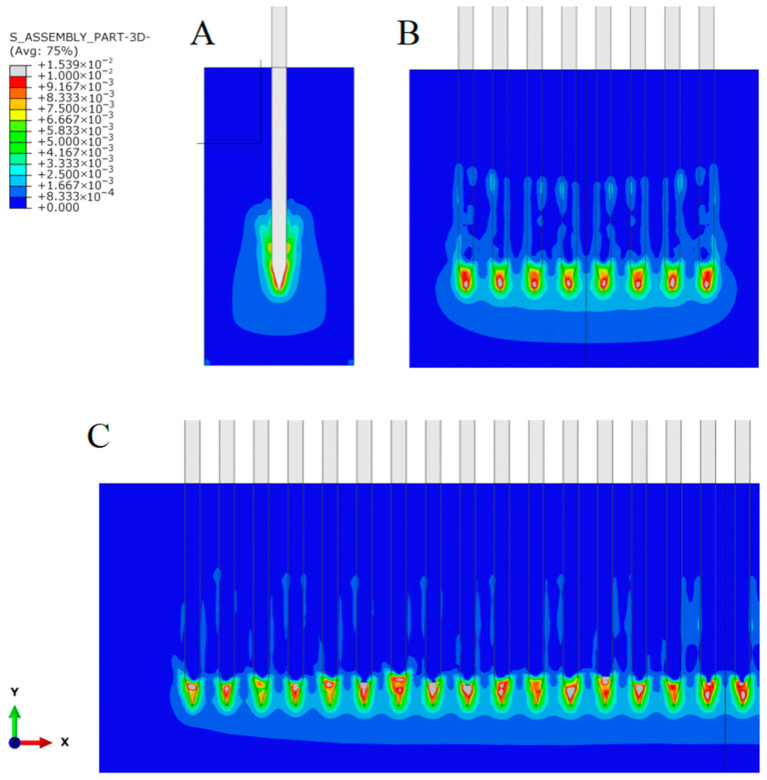
Stress in brain tissue at 1 mm implantation depth. (**A**) Single shank; (**B**) 8 shanks; (**C**) 32 shanks.

**Figure 8 micromachines-16-01010-f008:**
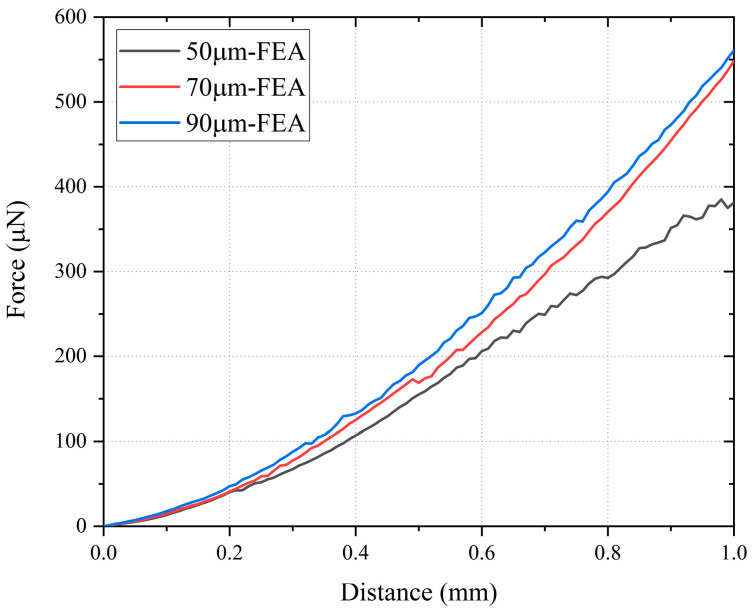
Insertion force versus depth for electrode shanks of different thicknesses. The insertion force exhibited comparable trends across all three electrode dimensions, although thicker shanks resulted in higher insertion forces.

**Figure 9 micromachines-16-01010-f009:**
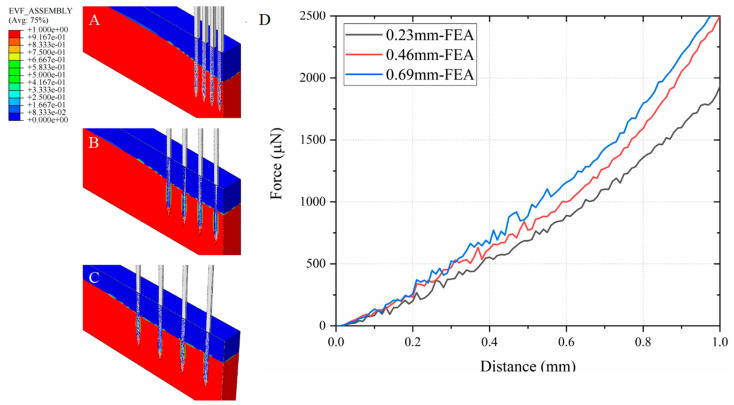
Simulated cross-sectional views of 8-shank array with center-to-center inter-shank spacing of (**A**) 0.23 mm, (**B**) 0.46 mm, and (**C**) 0.69 mm; (**D**) Insertion force versus depth for an 8-shank electrode with different shank spacing. The insertion force trends are consistent across arrays with varying spacing; however, larger inter-shank distances result in greater insertion forces due to reduced mechanical interaction between shanks.

**Figure 10 micromachines-16-01010-f010:**
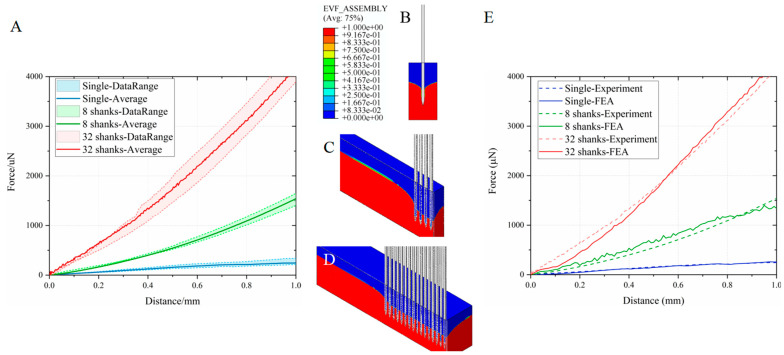
(**A**) Experimental results; Simulated cross-sectional views of (**B**) single-shank, (**C**) 8-shank, and (**D**) 32-shank electrode insertions, and (**E**) simulation-experiment comparison at 0.1 mm/s. The insertion force shows a similar depth-dependent trend as observed at 1 mm/s, but with reduced magnitude and less fluctuation. The simulation results maintain strong agreement with experimental measurements.

**Figure 11 micromachines-16-01010-f011:**
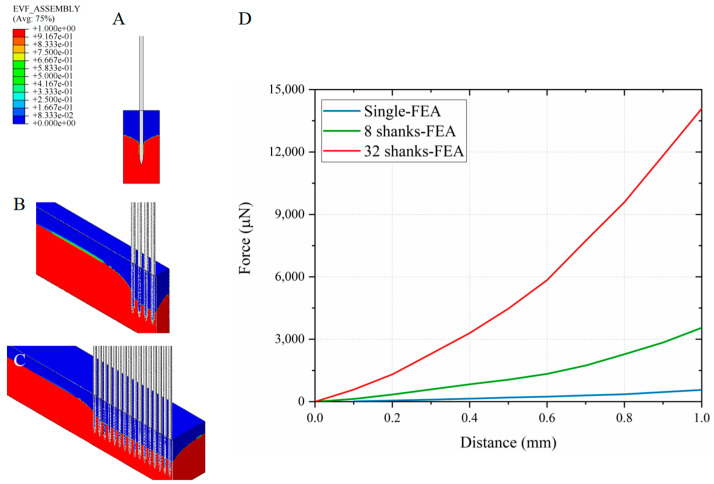
Simulated cross-sectional views of (**A**) single-shank, (**B**) 8-shank, and (**C**) 32-shank electrode insertions, and (**D**) simulation-experiment comparison at 10 mm/s.

**Figure 12 micromachines-16-01010-f012:**
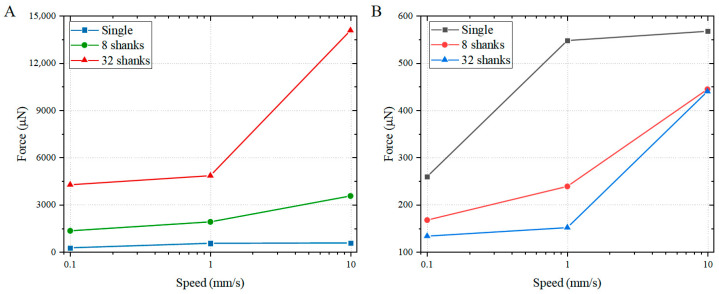
(**A**) Total insertion force and (**B**) average insertion force per shank under different insertion speeds. The insertion force increases with higher insertion speeds for all electrode configurations. Furthermore, the average insertion force per shank decreases as the number of shanks increases, which is consistent with the trend observed at the 1 mm/s insertion speed.

**Table 1 micromachines-16-01010-t001:** FEA model setup.

		Electrodes	Brain Tissue
Mesh Size	Interacting regions	0.02 mm	0.02 mm
	Other regions	0.1 mm	0.05 mm
Boundary Condition		Move 1 mm/s down total 1 mm along the shank	Fixing the motion of sides in the normal direction, Fixing the bottom
Material Properties		rigid body	Ogden Hyper Elastic
Domain Type		Lagrange	Euler

**Table 2 micromachines-16-01010-t002:** FEA model mesh number.

Number	Electrodes	Model Type	Eulerian Dimensions (mm)	Number of Elements
Case 1	Single	whole model	1 × 0.7 × 1.5	200,332
Case 2	8 shanks	axisymmetric model	4.6 × 3 × 1.5	1,091,396
Case 3	32 shanks	axisymmetric model	7.6 × 3 × 1.5	1,735,600

**Table 3 micromachines-16-01010-t003:** Properties of the FEA model.

	Ogden Hyperelastic Model		Density
Properties	*μ* (MPa)	*α*	ρ (kg∙mm^3^)
Values	0.001038	2.766	1.06 × 10^−3^

**Table 4 micromachines-16-01010-t004:** FEA Simulation Configurations for Various Cases.

Number	Electrodes	Shank Thicknesses (μm)	Shank Spacing (mm)	Insertion Speed (mm/s)
Case 1	Single	70	0.23	1
Case 2	8 shanks	70	0.23	1
Case 3	32 shanks	70	0.23	1
Case 4	Single	50	0.23	1
Case 5	Single	90	0.23	1
Case 6	8 shanks	70	0.46	1
Case 7	8 shanks	70	0.69	1
Case 8	Single	70	0.23	0.1
Case 9	8 shanks	70	0.23	0.1
Case 10	32 shanks	70	0.23	0.1
Case 11	Single	70	0.23	10
Case 12	8 shanks	70	0.23	10
Case 13	32 shanks	70	0.23	10

**Table 5 micromachines-16-01010-t005:** Average implantation force per shank for different electrode specifications.

Number	Electrodes	Total Implantation Force (μN, ±SD)	Average Implantation Force (μN)	Comparison with Single Shank
Case 1	1	538.6 ± 94.55	538.6	100%
Case 2	8 shanks	1879.4 ± 505.97	234.9	43.6%
Case 3	32 shanks	4792.7 ± 1444.86	149.8	27.8%

## Data Availability

The original contributions presented in this study are included in the article. Further inquiries can be directed to the corresponding author.

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
