# Peer review of "Mechanical Behavior Analysis of Neural Electrode Arrays Implantation in Brain Tissue"

_micromachines, 2025, doi:10.3390/mi16091010_

Round 1

Reviewer 1 Report

Comments and Suggestions for Authors

The current manuscript aims to conduct mechanical behavior analysis of neural electrode arrays implantation in brain tissue. Although the topic is interesting in its scientific field, there are some issues that require the authors’ attention to improve the quality of this particular manuscript before further consideration for publication in a high-quality journal “Micromachines”.

Specific comments:

  1. Why the authors select the Ogden hyperelastic model to describe the mechanical behavior of pig brain tissue? Please justify.
  2. Please provide a reasonable explanation for the observation that the stress impact range of 32 electrodes is smaller than that of 8 electrodes.
  3. The heterogeneity of brain tissue is well-known. Whether any differences in elastic modulus and viscoelastic properties can be detected across different brain regions (cortex and white matter)? Please clarify.
  4. How to obtain the coefficient of friction between brain tissue and the electrode surface? Please specify.
  5. The experiments only consider an implantation speed of 1 mm/s. Why? Please justify. Furthermore, it would be better to clarify the scientific meaning of this experimental parameter by examining the mechanical response and tissue damage under different implantation speed conditions.
  6. Regarding the introductory background, the authors mention that understanding the mechanical behavior of implanted neural electrodes is crucial for BCI development. Nevertheless, in my opinion, the understanding of the mechanical behavior of implants is not crucial for BCI development, but also important for many tissue engineering fields. In order to balance scientific viewpoints, the authors may consider the inclusion of the following relevant case study (DOI: 10.1016/j.actbio.2017.11.018) in the reference list to enrich the Introduction section and attract more attention from broad readers.

Author Response

Comments 1: Why the authors select the Ogden hyperelastic model to describe the mechanical behavior of pig brain tissue? Please justify.

Response 1: Thank you for pointing this out. The pronounced compliance and large-deformation behavior of brain tissue align well with the mechanical characteristics of hyperelastic materials. Furthermore, the use of a hyperelastic constitutive model for representing brain tissue is strongly supported by extensive previous research on its mechanical behavior [23-30]. We have added comprehensive explanations regarding this matter in the main text, see line 188-190, as suggested.

Comments 2: Please provide a reasonable explanation for the observation that the stress impact range of 32 electrodes is smaller than that of 8 electrodes.

Response 2: Your concern is highly valuable.

Figure 1 compares the deformation behavior of brain tissue under single-, 8-, and 32-electrode insertions under identical parameters (1 mm/s, depth of 1 mm). Consistent with the experimental observations in Figure 4 , it can be observed that as the number of electrode shanks increases, the degree of cortical surface tilt around each shank also increases.

During single-electrode insertion, the cortical surface forms a sharp-tipped groove centered around the shank. With an 8-electrode array, the tissue surface around each shank tilts toward the center of the array. In the case of the 32-electrode array, the cortical surface remains nearly horizontal across most central shanks, except for the outermost two or three shanks.

When the pushing force applied by the microelectrode causes tilting of the cortical surface, lateral compressive forces are imposed on the shank. A greater tilt angle results in more significant lateral loading, which leads to higher mechanical stress. Consequently, arrays with fewer shanks (e.g., single-shank) experience more pronounced stress concentrations, whereas the 32-electrode array exhibits a reduced stress impact range compared to the 8-electrode configuration.

Figure 1. Cross-sectional views along the electrode neutral plane after implantation to a depth of 1 mm for (A) single-shank, (B) 8-shank, and (C) 32-shank electrode configurations. Red and blue regions represent brain tissue and the overlying air, respectively. The extent of cortical surface deformation varies among configurations due to differences in shank count.

Comments 3: The heterogeneity of brain tissue is well-known. Whether any differences in elastic modulus and viscoelastic properties can be detected across different brain regions (cortex and white matter)? Please clarify.

Response 3:  Your concern is highly valuable. In our experimental study, the meninges were carefully removed from the cortical surface prior to electrode insertion. Moreover, the implantation depth of 1 mm was insufficient to penetrate beyond the most superficial layer of gray matter, thereby avoiding potential confounding effects arising from differences in mechanical properties across distinct brain regions. We have added explanations regarding this matter in the main text, see line 147-148, as suggested.

Comments 4: How to obtain the coefficient of friction between brain tissue and the electrode surface? Please specify.

Response 4:  Your concern is highly valuable. However, direct measurement of this coefficient is not feasible. Therefore, when establishing the simulation model, we referred to the friction coefficient parameters reported in [20] and made minor adjustments to ensure alignment between the simulation results and experimental data.

Comments 5: The experiments only consider an implantation speed of 1 mm/s. Why? Please justify. Furthermore, it would be better to clarify the scientific meaning of this experimental parameter by examining the mechanical response and tissue damage under different implantation speed conditions.

Response 5:  We have supplemented the study with experimental and simulation results at an insertion speed of 0.1 mm/s, along with simulated results at 10 mm/s. Due to equipment limitations, experimental measurements at 10 mm/s were not feasible. However, the model’s reliability has been rigorously validated through close agreement between simulations and experiments at both 0.1 mm/s and 1 mm/s. The simulation results shows a consistent increase in insertion force with higher insertion rates, as shown in figure 2. This finding significantly enriches our study by providing a broader mechanistic understanding of rate-dependent tissue-electrode interactions.

Figure2. (A) Total insertion force and (B) average insertion force per shank under different insertion speeds.The insertion force increases with higher insertion speeds for all electrode configurations. Furthermore, the average insertion force per shank decreases as the number of shanks increases, which is consistent with the trend observed at the 1 mm/s insertion speed.

We have added comprehensive explanations regarding this matter in the main text, see section 3.4 and line 382-392, as suggested.

Comments 6: Regarding the introductory background, the authors mention that understanding the mechanical behavior of implanted neural electrodes is crucial for BCI development. Nevertheless, in my opinion, the understanding of the mechanical behavior of implants is not crucial for BCI development, but also important for many tissue engineering fields. In order to balance scientific viewpoints, the authors may consider the inclusion of the following relevant case study (DOI: 10.1016/j.actbio.2017.11.018) in the reference list to enrich the Introduction section and attract more attention from broad readers.

Response 6:  We sincerely thank the reviewer for this suggestion. The recommended literature has been included in the revised manuscript as reference [29].

Reviewer 2 Report

Comments and Suggestions for Authors

The manuscript titled Mechanical Behavior Analysis of Neural Electrode Arrays Implantation in Brain Tissue by Xinyue Tan et al. addresses an important and relevant topic related to tissue engineering for brain-computer interfaces. However, further clarifications are needed before a decision can be made. Please see the comments below.

Titled: OK.

Abstract: The manuscript lacks information on the key performance indicators of the study, specifically the advantages of the proposed model and the novel aspects it introduces compared to the current state of the art.

Introduction:

In the context of this study, it would be valuable to review the current developments in the use of electrodes for neural applications. As a reference, I recommend the work by Neto et al., Transparent and Flexible Electrocorticography Electrode Arrays Based on Silver Nanowire Networks for Neural Recordings (Jun 25, 2021, ACS Applied Nano Materials, 4(6), pp. 5737-5747). This study focuses on developing high-performance, low-cost electrocorticography electrodes, and it would be insightful to evaluate how the present work advances or addresses this area.

For a broader perspective related to the core topic, see also Pereira et al., Flexible Active Crossbar Arrays Using Amorphous Oxide Semiconductor Technology Toward Artificial Neural Networks Hardware (Nov 2022, Advanced Electronic Materials, 8(11)).

Materials and Methods

How many electrodes were processed in total? How stable, reproducible, and reliable are the fabrication processes and the resulting structures? What is the measurement variability or error when evaluating arrays processed within the same batch but in different spatial regions, considering film uniformity and homogeneity? Additionally, what are the performance variations or errors observed across different fabrication runs?

Under what environmental conditions were the arrays and electrodes tested? Were any aging effects observed during testing? Did you perform bendability or flexibility tests? Such assessments are particularly relevant given the intended applications.

Results

Please begin by specifying the process technology employed, as well as the chosen design and architecture of the electrodes. Include details on the range of sizes and thicknesses used. Then, report the conductivity and sheet resistance values, along with an explanation of how these properties depend on the thicknesses and process parameters applied.

Additionally, identify the measurement errors or uncertainties associated with the values presented in the four tables.

Discussion

Discuss the influence of structure and layer thickness on the performance outcomes, beginning with an evaluation of the electrodes' resistance or conductivity, alongside the range of geometries chosen for the tested array.

Additionally, please elaborate further on the design model and architectural approach used for the array, highlighting how they impact overall performance and functionality.

Conclusions

The statement is too broad; it would be beneficial to specify the performance metrics achieved and how they contribute to advancing the current state of the art. Additionally, please clarify the specific advantages of using the proposed model and how it improves upon existing approaches.

Figures: Are OK.

Tables: Needs correction

References: require updating.

Author Response

Comments 1: Abstract: The manuscript lacks information on the key performance indicators of the study, specifically the advantages of the proposed model and the novel aspects it introduces compared to the current state of the art.

Response 1: We sincerely appreciate your suggestion. We agree that it is necessary to include the information you recommended in the abstract. We have carefully revised the abstract accordingly, with the modifications highlighted in red for your convenience.

Comments 2: Introduction:

In the context of this study, it would be valuable to review the current developments in the use of electrodes for neural applications. As a reference, I recommend the work by Neto et al., Transparent and Flexible Electrocorticography Electrode Arrays Based on Silver Nanowire Networks for Neural Recordings (Jun 25, 2021, ACS Applied Nano Materials, 4(6), pp. 5737-5747). This study focuses on developing high-performance, low-cost electrocorticography electrodes, and it would be insightful to evaluate how the present work advances or addresses this area.

For a broader perspective related to the core topic, see also Pereira et al., Flexible Active Crossbar Arrays Using Amorphous Oxide Semiconductor Technology Toward Artificial Neural Networks Hardware (Nov 2022, Advanced Electronic Materials, 8(11)).

Response 2: We sincerely thank the reviewer for these insightful recommendations. Both studies provide valuable context and have been incorporated into the revised manuscript as references [20, 21].

Comments 3: Materials and Methods: How many electrodes were processed in total? How stable, reproducible, and reliable are the fabrication processes and the resulting structures? What is the measurement variability or error when evaluating arrays processed within the same batch but in different spatial regions, considering film uniformity and homogeneity? Additionally, what are the performance variations or errors observed across different fabrication runs?

Response 3: We sincerely appreciate the reviewer’s insightful suggestion and meticulous attention to detail.

Indeed, the MEMS fabrication process used for our electrodes offers high stability and reproducibility. A single 8-inch wafer can yield dozens of electrodes in one batch. Electrodes fabricated on the same wafer within the same batch exhibit excellent consistency, with dimensional variations generally within 0.1 μm [1]. Even across different batches, the accuracy along the wafer axis remains within 0.7 μm, while variations normal to the wafer are typically below 0.4 μm [2].

It is worth noting that this study focuses specifically on simulating the acute mechanical interactions during electrode insertion into brain tissue, rather than the functional performance of the electrodes themselves. Given that the micron-scale dimensional variations inherent to the MEMS process are substantially smaller than the resolution of our mechanical model (0.02 mm, as determined by the FEA mesh size), we consider their influence on the simulation outcomes to be negligible.

Comments 4: Materials and Methods: Under what environmental conditions were the arrays and electrodes tested? Were any aging effects observed during testing? Did you perform bendability or flexibility tests? Such assessments are particularly relevant given the intended applications.

Response 4: Thank you for pointing this out. All implantation experiments were conducted at standard room temperature. Since this study focuses specifically on the acute phase of electrode insertion into brain tissue—a process lasting between 0.1 and 10 seconds depending on insertion speed—no aging effects of the tissue were observed. The electrodes used are rigid silicon-based devices, and therefore tests for bendability or flexibility were not performed. Throughout the experiments, no structural damage occurred in any of the electrodes.

Comments 5: Results: Please begin by specifying the process technology employed, as well as the chosen design and architecture of the electrodes. Include details on the range of sizes and thicknesses used. Then, report the conductivity and sheet resistance values, along with an explanation of how these properties depend on the thicknesses and process parameters applied.

Response 5: We sincerely appreciate the constructive feedback provided by the reviewer. After carefully considering existing literature on related studies, as well as our experimental procedures and simulation modeling methodology, we cautiously conclude that electrical parameters such as conductivity and sheet resistance have minimal impact on the mechanical processes under investigation in this study. That said, we fully acknowledge the importance of the reviewer's point regarding the significance of electrical performance and stability of electrodes. Our research team intends to pursue further dedicated studies on this aspect in future work. We would be grateful for the opportunity to engage in further discussion on this topic if possible.

Comments 6: Results: Additionally, identify the measurement errors or uncertainties associated with the values presented in the four tables.

Response 6: We fully agree that including error data alongside experimental results is essential for enhancing the rigor and reliability of the findings. Among the four tables in the manuscript, Tables 1–3 describe the structural, meshing, and material parameters of the simulation model, and therefore do not involve experimental errors. For Table 4, which summarizes the experimental measurements, we have now added error values based on the actual experimental outcomes. The following statement has also been added to the text , see lines 236-238.

Comments 7: Discussion: Discuss the influence of structure and layer thickness on the performance outcomes, beginning with an evaluation of the electrodes' resistance or conductivity, alongside the range of geometries chosen for the tested array.

Response 7: Thank you for pointing this out.

Taking the single-shank electrode as an example, we investigated the insertion forces of electrodes with different dimensions based on simulation results. As shown in Figure 1, the trend of insertion force versus depth is presented for shank thicknesses of 50 μm, 70 μm, and 90 μm. The results indicate that while the general trend of the insertion force curves remains consistent across different thicknesses, the magnitude of the force increases with greater shank thickness.

Figure A. Insertion force versus depth for electrode shanks of different thicknesses.

Additionally, please elaborate further on the design model and architectural approach used for the array, highlighting how they impact overall performance and functionality.

Due to the constraints of the planar fabrication process in MEMS technology, single-row multi-shank electrodes offer advantages in terms of production cost and consistency. The spacing between individual shanks was determined based on the structural dimensions of neurons [3] and existing studies on array-based neural electrodes. Using an 8-shank array as an example, we further investigated the insertion forces of arrays with different the center-to-center inter-shank spacing (0.23 mm, 0.46 mm, and 0.69 mm). The results indicate that the insertion force increases with larger inter- shank spacing, which can be attributed to reduced mechanical constraint between shanks and greater localized tissue deformation.

Figure B. Curve of insertion force versus depth for an 8-shank microelectrode with different inter-shank spacings.

For specific changes, see section 3.2 and lines 369-381:

Comments 8: Conclusions: The statement is too broad; it would be beneficial to specify the performance metrics achieved and how they contribute to advancing the current state of the art. Additionally, please clarify the specific advantages of using the proposed model and how it improves upon existing approaches.

Response 8: Thank you for pointing this out. In the Conclusion section, we have further emphasized the specific advantages of the proposed simulation model over existing approaches and elaborated on its significance in advancing neural electrode technology. The revised text is provided below, with modifications highlighted in red for clarity.

The revised conclusion is as follows:

“This study developed a validated FE model using the CEL method to simulate the complex dynamics of neural electrode implantation into brain tissue, specifically characterizing the biomechanical interactions and nonlinear tissue deformation mechanisms.

Experiments involving the implantation of single-shank and multi-shank electrodes into ex vivo porcine brain tissue demonstrated consistency between simulation results and actual measurements, thereby validating the model's accuracy. The results indicate a positive correlation between neural electrode insertion force and implantation depth. Furthermore, the average insertion force per electrode within a multi-electrode array decreases as the number of electrodes increases.

By providing a predictive simulation framework, this study helps reduce the need for extensive animal experiments, relying instead on carefully validated correlations from a limited set of porcine trials. These findings provide a basis for predicting the insertion forces of high-density neural electrode arrays and offer significant guidance for optimizing the structural design parameters of electrodes and the design of BCI implantation devices.”

1. Kim, G.H.; Kim, K.; Lee, E.; An, T.; Choi, W.; Lim, G.; Shin, J.H. Recent Progress on Microelectrodes in Neural Interfaces. Materials 2018, 11, 1995. https://doi.org/10.3390/ma11101995

2. Mirzazadeh, R.; Eftekhar Azam, S.; Mariani, S. Mechanical Characterization of Polysilicon MEMS: A Hybrid TMCMC/POD-Kriging Approach. Sensors 2018, 18, 1243. https://doi.org/10.3390/s18041243

3. Assaf Y. Imaging laminar structures in the gray matter with diffusion MRI. Neuroimage. 2019;197:677-688. doi:10.1016/j.neuroimage.2017.12.096

Round 2

Reviewer 1 Report

Comments and Suggestions for Authors

The revised version has adequately addressed most of the critiques raised by this reviewer and is now suitable for publication in "Micromachines".

Reviewer 2 Report

Comments and Suggestions for Authors

I agree with the set of corrections performed. The paper now is suitable for publication